# The variable presence of *Leptospira* in the environment; an epidemiological explanation based on serial analysis of water samples

**Janith Warnasekara**[1], **Shalka Srimantha**[1], **Indika Senavirathna**[1,2], **Chamila Kappagoda**[1], **Nirmani Farika**[1], **Achala Nawinna**[3], **Suneth Agampodi**[1,4]*

1 Faculty of Medicine and Allied Sciences, Department of Community Medicine, Rajarata University of Sri Lanka, Saliyapura, Sri Lanka, 2 Faculty of Medicine and Allied Sciences, Department of Biochemistry, Rajarata University of Sri Lanka, Saliyapura, Sri Lanka, 3 Faculty of Medicine and Allied Sciences, Rajarata University of Sri Lanka, Saliyapura, Sri Lanka, 4 Department of Internal Medicine, Section of Infectious Diseases, School of Medicine, Yale University, New Haven, Connecticut, United States of America

* sunethagampodi@yahoo.com

**Data Availability Statement:** All epidemiological data from the 3rd component have been deposited

## Abstract

Human leptospirosis involves the classic epidemiological triad (agent, host and environment); hence the investigations should include the knowledge on *Leptospira* within the animals and the environment. The objectives of this study are to explore the abundance of *Leptospira* in different climate zones of Sri Lanka and to describe the presence of *Leptospira* in the same water source at serial time points. First, water and soil samples were collected from different parts of Sri Lanka (Component-1); second, water sampling continued only in the dry zone (Component-2). Finally, serial water sampling from ten open wells was performed at five different time points (Component-3). Quantitative PCR of water and metagenomic sequencing of soil were performed to detect *Leptospira*. Three replicates for each sample were used for PCR testing, and positive result of two or more replicates was defined as 'strongly positive,' and one positive replicate was defined as positive. In the water and soil sample analysis in the whole country (Component-1), two out of 12 water sites were positive, and both were situated in the wet zone. Very small quantities of the genus *Leptospira* were detected by 16 amplicon analysis of soil in all 11 sites. In the dry zone water sample analysis (Component-2), only samples from 6 out of 26 sites were positive, of which one site was strongly positive. In the serial sample analysis (Component-3), Six, five, four, five, and six wells were positive in serial measurements. All wells were positive for at least one time point, while only one well was positive for all five time points. Proximity to the tank and greater distances from the main road were associated with strong positive results for *Leptospira* (P<0.05). The presence of *Leptospira* was not consistent, indicating the variable abundance of *Leptospira* in the natural environment. This intermittent nature of positivity could be explained by the repetitive contamination by animal urine.

at 10.5281/zenodo.4483786. All metagenomic data can be accessed through MG-RAST ID mgp-97439.

**Funding:** SA: JW and IS are partially supported by US Public Service Grant U19AI115658. The sample analysis was funded through the Faculty of Medicine and Allied Sciences, Rajarata University of Sri Lanka annual publication award received by the first and last authors. All the other costs are self-funded. The funders had no role in study design, data collection and analysis, decision to publish, or preparation of the manuscript.

**Competing interests:** The authors have declared that no competing interests exist.

## Introduction

Integrating the knowledge on human, animal, and environmental health is essential in controlling and predicting zoonotic diseases. While investigations on animal and human interfaces are increasing, greater incorporation of environmental and ecosystem components is highlighted as a missing link in the One Health approach [1]. Leptospirosis, a globally widespread and neglected tropical disease, also lacks adequate investigations linking animal and environmental factors to human infection. Various definitive and intermediate hosts, such as livestock, domestic pets, and wild or feral animals, harbour *Leptospira* in their proximal convoluted tubules of renal nephrons and excrete *Leptospira* via urine [2]. These excreted *Leptospira* enter the human body through abrasions of the skin, mucus membranes, or conjunctiva and cause leptospirosis [3]. In addition, *Leptospira* has acquired different mechanisms for adaptation to different environments [4].

Leptospirosis is a zoonotic disease transmitted mainly by mammals. People who directly contact animals or animal products and reside or work close to animal habitats are considered at risk for infection [2]. Hunters [5], sewer workers [6], butchers [7], veterinarians [8] and dairy farmers [9] are reported as major risk groups for the disease through direct exposure to animals, whereas farmers [10] and mine workers [11,12] have exposure via contaminated water sources. Studies have shown that contaminated water is a major source of disease transmission, as the disease is associated with floods, rainfall, and recreational activities in water [13,14]. Unlike direct exposure, *Leptospira* has to enter the host within a short period after being shed into the environment or survive in water for a considerable period of time to cause disease by water contamination. Evidence suggests that *Leptospira* can survive in water for several days to more than one year [15]. Additionally, it has been revealed that *Leptospira* can cause infection in susceptible individuals even after prolonged starvation of the pathogen [15]. However, all the people who are exposed to contaminated water do not develop the infection. This phenomenon warrants further exploration of the mechanism of *Leptospira* transmission.

Sri Lanka is a leptospirosis hotspot [16,17], and the disease causes significant morbidity and mortality despite its underestimation in Sri Lanka [18,19]. The major modes of exposure to leptospirosis in Sri Lanka are paddy farming and working in gem mines [20]. This finding indicates that indirect exposure through water sources is more common in Sri Lanka than direct exposure to animals. Furthermore, evidence suggests that the infecting species and clinical patterns of leptospirosis vary among geographical locations in the country [21]. This indicates that the natural survival of *Leptospira* could vary among those areas. There are three major climate zones in Sri Lanka: the wet zone, the dry zone, and the intermediate zone [22]. On average, the wet zone receives high rainfall and frequently reports more leptospirosis cases than other zones, while the dry zone reports leptospirosis cases predominantly during the rainy season [23]. Livestock, farming practices, and wildlife are also different among these zones. All these factors may lead to varying degrees of *Leptospira* survival in natural water sources. The objectives of this study are to explore the presence of *Leptospira* in the environment around human habitats where leptospirosis cases are reported in different climate zones and to perform a time series evaluation of the abundance of *Leptospira* in natural water sources, the main human-animal interface of disease transmission.

## Materials and methods

### Study design and setting

This study included three major components of environmental sample collections, as illustrated in Fig 1. Firstly, island-wide (including most parts of the island) water and soil

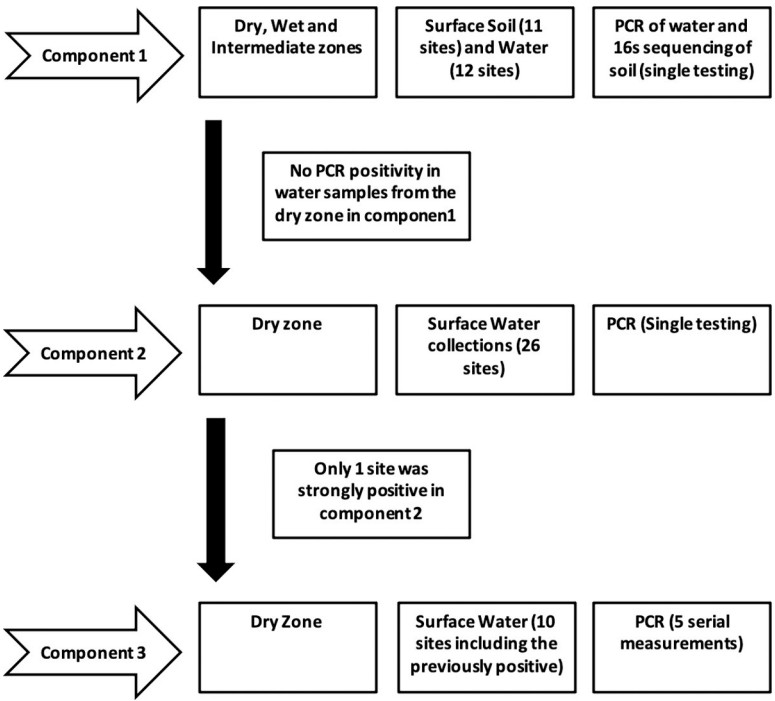

**Fig 1. Study flow chart of sampling process of the three components of the study.**

sample collection; secondly, dry zone water sample collection and thirdly serial sampling of water samples from ten open wells at five different time points. The first component (the whole country sample collection) had two subcomponents: water sample analysis and soil sample analysis. (Fig 1) For the water samples, water sources were selected purposefully based on the probable sites of contamination of diagnosed leptospirosis patients. Sampling was conducted at 12 sites representing all three climatic zones: dry, wet, and intermediate. As shown in Table 1, soil samples were collected from the same sites where the water samples had been collected. Site Katugasthota (Fig 2) was a deep canal where we collected only water due to the practical inconvenience of soil sample collection. All sites were selected based on the probable exposure history of confirmed patients with leptospirosis.

The second component included sample collection in the dry zone. For the second component, water collection sites were selected purposefully considering the possibility of daily human contacts (Fig 3). The Public Health Research Laboratory of the Faculty of Medicine and Allied Sciences, Rajarata University of Sri Lanka (FMAS_RUSL), was selected as the central point. Twenty-six nearby sites were selected considering a higher possibility of human being contact. The dry zone was selected over the wet zone for the second component, considering the lack of positive sites in the dry zone in the first component of the study.

For the third component, the strongly positive site (Site number 9—Open well) of the previous component (dry zone water collection) was selected as the central point. Then, another nine wells (altogether ten wells) from the close locality of this selected well were selected for the serial sampling of the third component. Thus, serial sampling was conducted at 2-week intervals four times, and the fifth sampling was conducted four weeks after the fourth sampling (Fig 4).

**Table 1. Presence of pathogenic *Leptospira* in environmental surface water samples and Genus *Leptospira* in soil samples from dry, wet and intermediate zones in Sri Lanka.**

| Sample Name | Zone | Site description | Water PCR | Soil Sample number (PA950_)# | Soil RA* (per 100,000) | Soil MG-RAST ID |
|---|---|---|---|---|---|---|
| Anuradhapura 1 | Dry | Bank of a tank | Neg | AP1_1 | 13.6 | mgm4919260.3 |
| | | | | AP2_1 | 48.6 | mgm4919257.3 |
| | | | | AP1_F06_23_01_2019 | 12.7 | mgm4919246.3 |
| Anuradhapura 2 | Dry | Bank of a tank | Neg | AP3_F05_23_01_2019 | 5.8 | mgm4919241.3 |
| | | | | AP1_2 | 4.6 | mgm4919261.3 |
| | | | | AP2_F07_23_01_2019 | 4.4 | mgm4919256.3 |
| | | | | AP2_F04_23_01_2019 | 20.1 | mgm4919245.3 |
| Ibbankatuwa | Imdt+ | Paddy field | Neg | IK5_1_F07 | 35.5 | mgm4919243.3 |
| | | | | IK1_1_V341F_10 | 0 | mgm4919254.3 |
| Katugasthota | Wet | Water Canal | Neg | Soils samples were not taken | | |
| Mawanella | Wet | Abandoned paddy field | Pos | MVN3_1_F02_31_01_2019 | 33.5 | mgm4919249.3 |
| | | | | MVN4_1_F03_31_01_2019 | 43.5 | mgm4919262.3 |
| | | | | MVN1_1_F01_31_01_2019 | 9.6 | mgm4919269.3 |
| Rathnapura 1 | Wet | Gem mine | Neg | RT4_1_F10 | 65.0 | mgm4919265.3 |
| | | | | RT2_1_F09 | 105.6 | mgm4919244.3 |
| | | | | RT1_1_F08 | 77.0 | mgm4919267.3 |
| Rathnapura 2 | Wet | Water canal | Neg | S25 | 31.4 | mgm4919258.3 |
| | | | | S24 | 29.7 | mgm4919255.3 |
| | | | | F07 | 26.7 | mgm4919242.3 |
| Galle 1 | Wet | Water Pit | Neg | S21_F02_new | 11.3 | mgm4919270.3 |
| Galle 2 | Wet | Paddy field | Neg | S22_F03_new | 8.8 | mgm4919266.3 |
| | | | | S20_F01_new | 8.2 | mgm4919252.3 |
| | | | | S23 | 15.3 | mgm4919264.3 |
| Mathara | Wet | Paddy field | Pos | F08 | 2.7 | mgm4919251.3 |
| | | | | F09 | 11.0 | mgm4919250.3 |
| | | | | F10 | 24.7 | mgm4919248.3 |
| Gampaha | Wet | Water Canal | Neg | GP4_1_V341F_09 | 6.0 | mgm4919259.3 |
| | | | | GP1_1_V341F_08 | 13.8 | mgm4919247.3 |
| Kuliyapitiya | Imdt | Paddy field | Neg | KU1 | 0.7 | mgm4919253.3 |
| | | | | KU2 | 0.7 | mgm4919263.3 |
| | | | | KU4 | 2.7 | mgm4919268.3 |

+Intermediate Zone

*Relative Abundance

#PA950_ precedes all the sample names, RA-Relative abundance.

## Sample collection and transport

Four water samples were collected from each site, and a one-meter gap was maintained between the sample collection locations within the sites. Ten millilitres of water was collected into a sterile 15 ml Falcon tube using a clean plastic container, and the lid was closed immediately. Samples were transported on ice packs to the Public Health Research Laboratory of the Faculty of Medicine and Allied Sciences, Rajarata University of Sri Lanka, within 48 hours of collection. Soil samples were collected only in the whole country sample collection. (Component 1- Fig 1) Four samples were collected from each site for the soil samples, maintaining a one-metre gap between the sample collection points within the site. Samples were collected

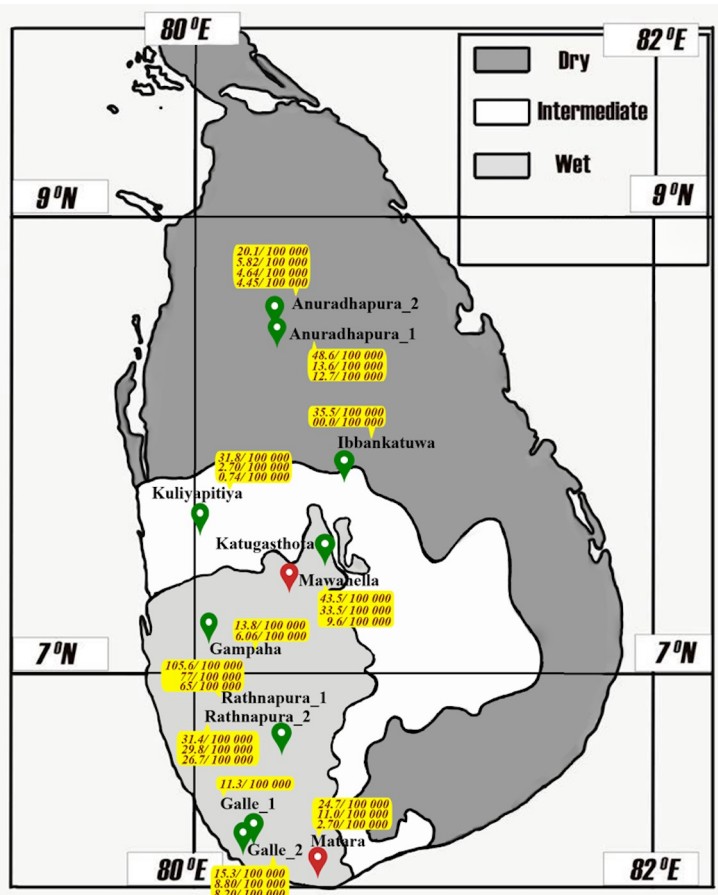

**Fig 2. PCR results of pathogenic *Leptospira* in environmental surface water samples and the relative abundance of the genus *Leptospira* in the soil microbiome from dry, wet and intermediate zones in Sri Lanka.** (Red–PCR-positive water, green–PCR-negative water, values within yellow background–relative abundance of the genus *Leptospira* in soil).

into a clean container using a clean spoon. The samples were transported the same way as the water samples. Eighteen samples from the wet zone (8 sites), five samples from the intermediate zone (2 sites), and seven samples from the dry zone (2 sites) were sent for metagenomic analysis.

## Sample processing, DNA extraction, and PCR testing

There is no optimized best method for concentrating *Leptospira* from water samples [24,25]. We found that a two-step protocol suggested by Paula et al. to concentrate *Leptospira* from urine produced better results than the available protocols for water [26]. Therefore, centrifugation was conducted in two steps. First, samples were centrifuged at 3000 rpm for 5 minutes to allow the debris to be deposited in the bottom of the tube. Second, each supernatant was transferred to two microcentrifuge tubes (1.5 mL) and centrifuged at 15000 rpm for 10 minutes, and the supernatants were discarded, and the deposit was used for the further steps. Samples collected from the same sites were pooled for the extraction of DNA. According to the manufacturer's instructions, DNA was extracted using a QIAamp DNA Blood Mini Kit (Qiagen, USA). This extraction kit is validated for environmental water by Julie Vein et al. [27] The pathogen-specific rRNA 16S-1 primer pair used for this study was described in a previous

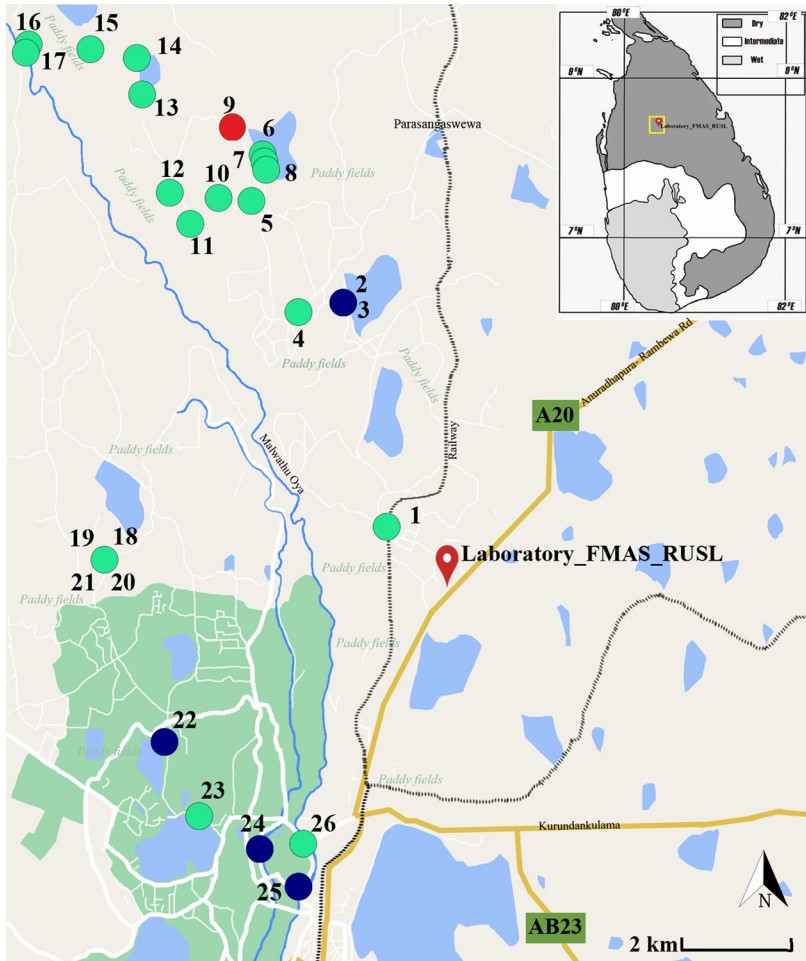

**Fig 3. Distribution of sample collection sites and the presence of pathogenic *Leptospira* in water samples in the dry zone, Sri Lanka.** (Green–Negative, Blue–Positive, Red–Strongly positive).

study as 16–1 F 5'-GCG TAG GCG GAC ATG TAA GT-3' and 16–1 R 5'-AAT CCC GTT CAC TAC CCA CG-3' [28]. qPCR was performed using the CFX96 real-time PCR detection system (Bio-Rad, US) with the following thermal cycle conditions: 95˚C for 5 minutes, 45 cycles of [94˚C for 30 s, 60˚C for 30 s], followed by melt curve generation from 65˚C to 90˚C performed at an increment of 0.5˚C per cycle. The PCR volumes were as follows. For each reaction well, 10 μL of SYBR Green Fast Mix (Quantabio, USA), 5 μL of DNA template, and 0.02 μL of each diluted forward and reverse primer were added. The total reaction volume was adjusted to 20 μL by adding PCR-grade water. The final concentration of each primer was 0.1μM.

## Definition of PCR positivity

As mentioned above, samples from the same site were pooled, and the pooled sample was used for DNA extraction. Three replicates from each pooled sample were included in PCR analysis. A positive curve with melting temperature was considered a positive replicate. If only one replicate was positive, the sample was considered positive. If two or more replicates were positive, the site was considered strongly positive.

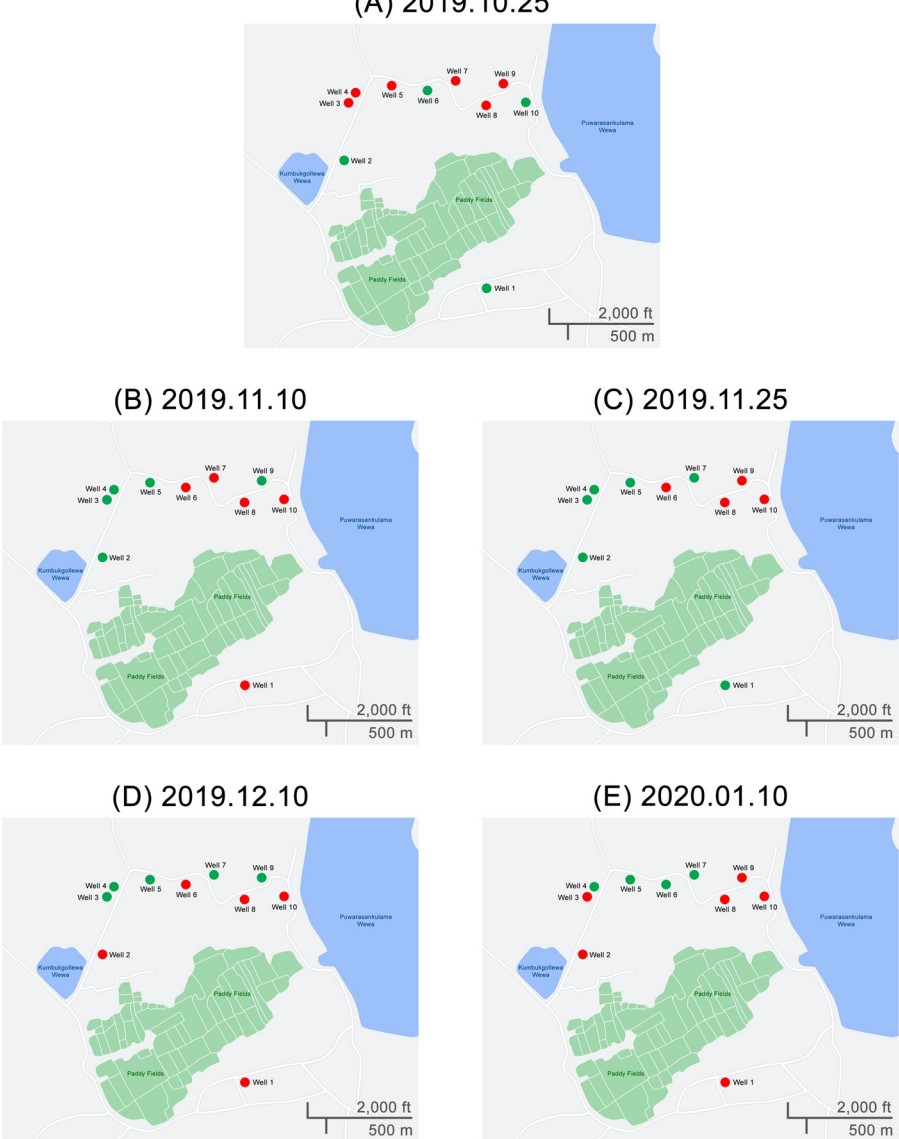

**Fig 4. Presence of Pathogenic *Leptospira* spp. in 10 open wells from the dry zone, Sri Lanka, over a period of 10 weeks.** Green-negative, red-positive (Well 9 is the strongly positive well in the second component).

## 16. S Amplicon sequencing of soil

Next-generation sequencing was performed at a commercial facility [29]. Bacterial *16S rRNA* amplicon sequencing was performed by Ion Torrent to detect the microbiota of the soil. According to the manufacturer's instructions, DNA extraction was performed using a soil-specific QIAAmp® DNA Mini Kit (Qiagen). The *16S rRNA* gene V1-V2 region was used first to confirm the presence of bacteria. Then, PCR amplification was performed in a 25 μl mixture containing 12.5 μl of Platinum® PCR Supermix (Invitrogen), 12.5 μM each primer and 3.75 μl of template DNA. Sequencing of the final libraries and template preparation was performed using the PGM™ Hi-Q™ OT2 Kit (Thermo Fisher Scientific) and Ion PGM™ Hi-Q™ Sequencing Kit (Thermo Fisher Scientific) according to the manufacturer's instructions. Barcoded bacterial libraries were multiplexed on a single chip in a 400 bp run to obtain

**Table 2. Local and environmental risk factors for 'well' positivity.**

| Significant feature | Subcategory | Mean positivity | t value | p-value |
|---|---|---|---|---|
| **Local factors of the well** | | | | |
| Built wall above ground level | Yes | 2.29 | t = 0.79 | p = 0.44 |
| | No | 3.00 | | |
| Frequency of well use per day | 1 or less | 2.60 | t = 0.23 | p = 0.82 |
| | 2 or more | 2.40 | | |
| Shed | Open | 2.20 | t = 0.72 | p = 0.48 |
| | Covered | 2.80 | | |
| Nutrification | Yes | 2.00 | t = 0.98 | p = 0.35 |
| | No | 2.86 | | |
| **Environmental factors of the well** | | | | |
| Distance from ground to water level | < 1 m | 2.25 | t = 0.48 | p = 0.64 |
| | >1 m | 2.67 | | |
| Distance to forest | < 2 km | 2.83 | t = 1.01 | p = 0.33 |
| | >2 km | 2.00 | | |
| Distance to paddy field | < 50 m | 2.50 | t = 0.00 | p = 1.00 |
| | >50 m | 2.50 | | |
| Distance to Chena | < 50 m | 2.00 | t = 1.2 | p = 0.23 |
| | >50 m | 3.00 | | |
| Distance to nearest water tank | < 800 m | 3.33 | t = 3.19 | p = 0.01 |
| | >800 m | 1.50 | | |
| Distance from main road | < 1 km* | 1.80 | t = 2.53 | p = 0.03 |
| | >1 km | 3.40 | | |

*less than the distance

#more than the distance.

sequencing data. Finally, bioinformatic analysis was performed by the investigators by uploading raw fastq data to Metagenome Rapid Annotation using the Subsystem Technology (MG-RAST) server [30].

## Data analysis

The total number of times that a well was positive out of five measurements was considered the dependent variable. Wells were categorized into two groups based on the presence or absence of the risk factors shown in Table 2. Two-sample t-tests were used to compare the mean positivity between risk factors present and absent wells. A p-value less than 0.05 was considered significant.

## Results

### Presence of pathogenic *Leptospira spp.* in water and genus *Leptospira* in soil from dry, wet and intermediate zones in Sri Lanka

Water and soil sample collection was performed at 12 sites in nine districts representing five out of the nine provinces of Sri Lanka (Table 1). Of the water samples tested from 12 sites, only the samples from Mawanella (an abandoned paddy field) and Mathara (a paddy field) tested positive for pathogenic *Leptospira* (Fig 2) in qPCR. Both sites were situated in the wet zone.

The *16S rRNA* amplicon sequencing data were analysed from eleven sites. Taxonomy assigned based on the RefSeq database via MG-RAST showed that in the soil microbiome, the

relative abundance of the genus *Leptospira* was minute compared to that of other organisms. The highest relative abundance (105.6) was reported from one of the samples of site Rathnapura 1 of the wet zone (Fig 2). Thus, these soil samples can comprise both pathogenic and non-pathogenic species, and non-pathogenic species are ubiquitous.

### Presence of pathogenic *Leptospira* in water samples from dry zone

The 26 sites included in the second component (dry zone sample collection) included water samples from large human-made irrigation tanks/lakes (n = 6), paddy fields (n = 6), rainwater collections (n = 4), rivers/natural water streams (n = 4), natural water pools (n = 2), water canals (n = 2) and wells (n = 2). Of these, a single site was strongly positive for *Leptospira*, while sites 2, 3, 22, 24, and 25 were positive (Fig 3). The strongly positive site 9 was a well from which water was used for agriculture and household activities but not for drinking.

### Presence of pathogenic *Leptospira* spp. in open wells in different time points

Fig 4A–4E show the PCR results of five serial samplings of the ten wells selected for the third component of the study. All the wells were positive in at least one of the five serial measurements. A minimum of four wells was positive at any time.

Table 2 summarizes the local and environmental factors associated with the number of times that each well was positive. All the wells shared similar characteristics, while the positivity was higher in the wells situated close to the water tank (lake) and away from the main road.

Fig 5 shows the association between well positivity and distance from the nearest water tank in kilometres. It clearly shows that when the distance from the water tank is reduced, the number of times that the well is positive increases.

## Discussion

To understand the transmission of human leptospirosis, focusing on the environment as the animal-human interface is required. However, cross-sectional studies with a single time point

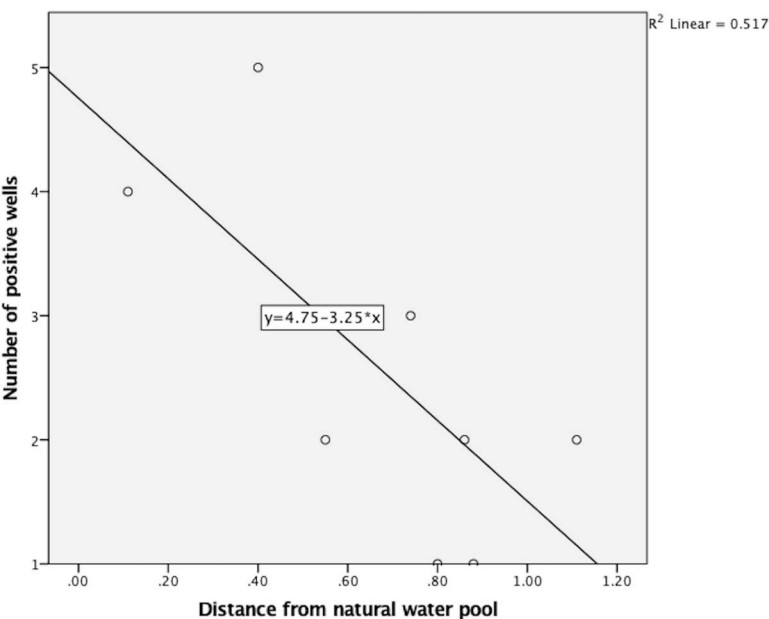

**Fig 5. Association between well positivity and distance from the nearest water pool (km).**

description of environmental contamination only partially explain the actual risks and transmission pattern of the disease. Therefore, in this study, we aimed to describe *Leptospira* in soil and water together with a serial sampling of water sources to describe the existence of *Leptospira* in the natural environment.

Although we are not intending to compare the results of this study with the clinical data, the finding that two sites in the wet zone were positive while none of the sites in the dry and intermediate zones were positive for *Leptospira* is compatible with the reported incidence of leptospirosis, as the wet zone reports nearly two times the number of cases compared to that in the dry zone (Fig 2) [31]. As the environment of the wet zone is favourable for the growth and survival of *Leptospira*, the probability of detecting the organism in samples is expected to be higher. The observed difference between wet and dry zones could also be due to the diversity of *Leptospira* in different geographical areas, as described previously [32]. It has been shown that *Leptospira* can survive in vitro as well as in the natural environment through biofilm formation with the environmental microbiota. Therefore, *Leptospira* can survive even in nutrient-free environments [33,34]. On the other hand, the nutrients required to survive diverse *Leptospira* in the two climatic zones could be different. Further studies are needed to explain the differences we observed. Nutrient availability could be the main reason for the observed diversities of water samples between the climate zones. In addition, the diversity of the soil microbiome may be a contributory factor to the differences we observed, as shown in the *16S* amplicon sequencing data of the soil samples tested at the same sites [35]. Species- or strain-specific differences in the natural survival of *Leptospira*, with a specific focus on geographical, environmental, and climatic factors, need further exploration [15].

An emerging hypothesis is that virulent *Leptospira* survive in the soil for a long period and come to the surface when the soil is washed away during the rainy season [15,36,37]. Therefore, the probability of detecting *Leptospira* could be higher in the wet zone due to its surface wetness throughout the year. In the dry zone, relatively low rainfall is received for a short period of time [22]. During the non-rainy season, the lands become completely dry, making them unfavourable for the growth and survival of *Leptospira*. Therefore, leptospirosis outbreaks occur predominantly during the rainy season in the dry zone. Although this study was conducted during the rainy season, the presence of *Leptospira* in the dry zone was still less frequent than that in the wet zone. With *16S* amplicon sequencing, the genus *Leptospira* was detected from the soil samples at all 11 sites. The negativity of the water samples but the positivity of the soil samples could support the emerging hypothesis of a higher abundance of *Leptospira* in soil than in water [15,36,37]. However, the genus *Leptospira* detected in soil samples includes pathogenic and non-pathogenic species, although the primer used for PCR of water samples specifically targeted pathogenic *Leptospira*. Nevertheless, the observed difference may have been due to the different techniques we performed for soil and water, which is a major consideration and limitation when interpreting the findings. To transmit *Leptospira* from the environment to humans, a satisfactory concentration of *Leptospira* is needed in the environment when contact with the potential host. Therefore, the very low concentration of *Leptospira* in both soil and water in random samples may explain the necessity of acute exposure to an environment containing *Leptospira* for human transmission. This highlights the importance of exposure to *Leptospira*-containing animal urine shortly after excretion to consider the environment infectious for human transmission at the time of contact between the environment and humans. This hypothesis is further supported by the serial water sample analysis as 'well positivity' is higher when the distance from the main road increases. This is further explained below in the discussion. Studies on required minimal *Leptospira* concentration in the environment could be new research area to be explored.

In the second component (dry zone water sample analysis of 26 sites), there was only one strong positive result, which was in an uncovered well. This finding is compatible with some of the previous knowledge stating that the isolation rate of *Leptospira* is higher in stagnated water than in running water [15]. This finding further confirms the variability of *Leptospira* abundance in the dry zone. Only a single well provided positive results throughout the serial testing (third component). Although the evidence suggests that some of the species can survive up to one year in the natural environment, our observation indicates a short lifecycle for *Leptospira* in the selected water source [15]. However, the theoretical possibility of 'non-even distribution of *Leptospira* in well water could have led to the non-inclusion of *Leptospira* in the obtained water sample, incorrectly leading to the finding of a short life span of the organism. This preliminary observation indicates that in-depth exploration of water's physical and chemical qualities with serial samples is required to understand *Leptospira* survival in water. However, recontamination between samplings will be a major confounding effect and must be avoided in future studies. With this observation, distance from main roads was also associated with positivity, which supports the hypothesis of more frequent contamination by feral animals. A higher positive rate closer to water pools indicates an association of leptospirosis with aquatic environments. This is compatible with the findings of a systematic review published by Mwanajaa et al., where most water-related activities were identified as significant risk factors for leptospirosis [2].

The intermittent nature of positivity could be better explained by repetitive contamination by the contaminated urine of animals. The ecological system in Sri Lanka allows domesticated animals, livestock, and feral animals to be mixed frequently, and in the study area, numerous rodent species are abandoned. This is compatible with the findings highlighted by Vincent et al. in 2019, where they stated that repetitive exposure could be the main risk factor for *Leptospira* infection [38]. Further, the authors highlighted the difficulty of obtaining definitive proof regarding the source of contamination of the environment through field studies. As the source of infection is best detected by investigating reservoirs, further studies targeting animals, humans, and the environment and their interactions are important to prevent the disease.

The finding of this study could be used to enhance the epidemiological triad of leptospirosis. We emphasize the variable presence of *Leptospira* in the environment as a major component of this epidemiological triad (Fig 6) to explain why all people who share the same exposure do not get the infection.

Prophylactic therapy with doxycycline for humans is the only preventive method recommended in some countries, despite the lack of evidence to support its use [39]. This hypothetical model provides an understanding of the different opportunities to search for new preventive methods for leptospirosis. Epidemiological studies alone cannot decide specific prophylactic measures as a replacement for doxycycline. Therefore, novel studies on the prevention of leptospirosis via changes in soil abundance, biological prevention, prevention through environmental toxins, avoiding entry to the body, and changing the host response by immune modulation can be explored as new avenues of leptospirosis prevention research in the future.

## Limitations

The sensitivity of PCR depends on Leptospiraemia [40]. As *Leptospira* is diluted in water, there is a high probability of failing to detect the existing *Leptospira* from the source of water collection. We used a 2-step centrifugation protocol to concentrate *Leptospira*. Although it was an optimized procedure, there was a probability of losing a considerable number of *Leptospira*

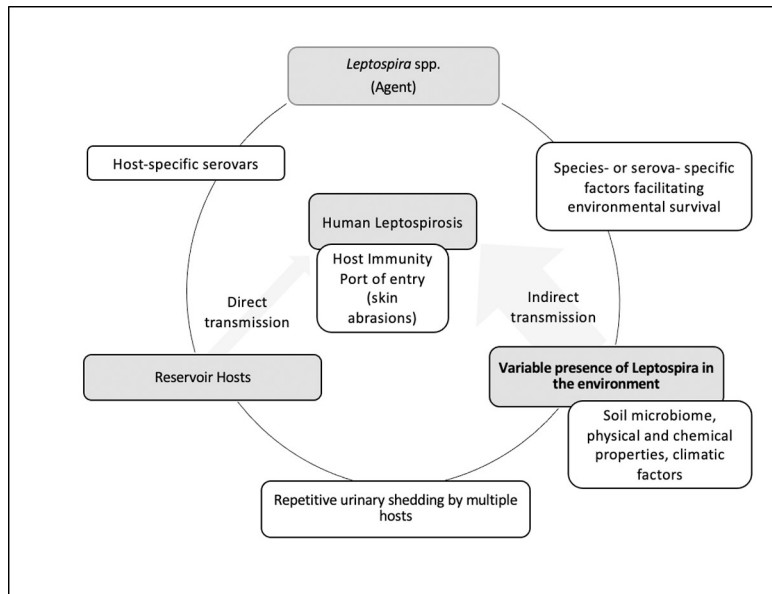

**Fig 6. The epidemiological triad for *Leptospira* to illustrate the effect of variable presence of *Leptospira* for the disease transmission.**

in the pellet of the 1$^{st}$ centrifugation step. Also, the DNA extraction kit we used was a tissue and blood kit. This was optimized for environmental water samples by Julie Vein et al. [27]. However, evidence suggests that the use of a water-specific DNA extraction kit enhances the DNA yield [41]. The metagenomic analysis is highly dependent on the database used. Therefore, we accept that there is a probability of missing species of the genus *Leptospira* that are not present in the MG-RAST database. Direct comparison of the abundances of *Leptospira* in water and soil could not be performed, as two different analysis techniques were used. Although the PCR was negative, there was a theoretical possibility of *Leptospira* being present in place with the same water collection other than the site of sample collection.

## Acknowledgments

We acknowledge all technical staff who worked in the Public Health Research Laboratory of the Faculty of Medicine and Allied Sciences, Rajarata University of Sri Lanka; Dr. Gihan Rathnayaka and Dr. Hasantha Banduwardana and Dr. Ishanka Amerasinghe for graphic design; and all the owners of the wells in the 3$^{rd}$ component of the study.

## Author Contributions

**Conceptualization:** Janith Warnasekara, Suneth Agampodi.

**Data curation:** Janith Warnasekara, Nirmani Farika, Achala Nawinna.

**Formal analysis:** Janith Warnasekara.

**Funding acquisition:** Janith Warnasekara, Suneth Agampodi.

**Investigation:** Janith Warnasekara, Shalka Srimantha, Chamila Kappagoda, Nirmani Farika, Achala Nawinna.

**Methodology:** Janith Warnasekara, Suneth Agampodi.

**Project administration:** Janith Warnasekara.

**Resources:** Suneth Agampodi.

**Software:** Janith Warnasekara.

**Supervision:** Suneth Agampodi.

**Validation:** Janith Warnasekara, Shalka Srimantha, Indika Senavirathna, Chamila Kappagoda.

**Visualization:** Janith Warnasekara.

**Writing – original draft:** Janith Warnasekara, Shalka Srimantha.

**Writing – review & editing:** Indika Senavirathna, Suneth Agampodi.

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
