## [Decision Letter · Decision Letter 0]

29 Oct 2021

PONE-D-21-25422Is the presence of Leptospira in the environment merely random? An epidemiological explanation based on serial analysis of water samplesPLOS ONE

Dear Dr. Agampodi,

Thank you for submitting your manuscript to PLOS ONE. After careful consideration, we feel that this manuscript has merit but does not fully meet PLOS ONE’s publication criteria as it currently stands. Therefore, we invite you to submit a revised version of the manuscript that addresses the points raised during the review process. While all reviewers thought that the study has merit, many aspects of the study were not described well enough for the reader to understand how the data was acquired, the rigor applied to statistics, and how the data was interpreted.  Thus, for this manuscript to be reconsidered for publication, it will be important for the authors to address ALL the comments brought forward by the reviewers.

 ==

We look forward to receiving your revised manuscript.

Kind regards,

R. Mark Wooten, Ph.D.

Academic Editor

PLOS ONE

2. "Funding Information and Financial Disclosure sections do not match:

We note that the grant information you provided in the ‘Funding Information’ and ‘Financial Disclosure’ sections do not match.

“JW and IS are partially supported by US Public Service Grant U19AI115658. The sample analysis was funded through the Faculty of Medicine and Allied Sciences annual publication award received by the first and last authors. All the other costs are self-funded.**”**

 “SA: JW and IS are partially supported by US Public Service Grant U19AI115658. The sample analysis was funded through the Faculty of Medicine and Allied Sciences annual publication award received by the first and last authors. All the other costs are self-funded. The funders had no role in study design, data collection and analysis, decision to publish, or preparation of the manuscript.”

5. We note that Figures 2,3 and 4 in your submission contain [map/satellite] images which may be copyrighted. All PLOS content is published under the Creative Commons Attribution License (CC BY 4.0), which means that the manuscript, images, and Supporting Information files will be freely available online, and any third party is permitted to access, download, copy, distribute, and use these materials in any way, even commercially, with proper attribution. For these reasons, we cannot publish previously copyrighted maps or satellite images created using proprietary data, such as Google software (Google Maps, Street View, and Earth). For more information, see our copyright guidelines: http://journals.plos.org/plosone/s/licenses-and-copyright.

a. You may seek permission from the original copyright holder of Figures 2, 3 and 4 to publish the content specifically under the CC BY 4.0 license. 

Reviewers' comments:

Reviewer's Responses to Questions

**Comments to the Author**

1. Is the manuscript technically sound, and do the data support the conclusions?

Reviewer #1: Partly

Reviewer #2: Yes

2. Has the statistical analysis been performed appropriately and rigorously? 

Reviewer #1: N/A

Reviewer #2: Yes

3. Have the authors made all data underlying the findings in their manuscript fully available?

Reviewer #1: Yes

Reviewer #2: Yes

4. Is the manuscript presented in an intelligible fashion and written in standard English?

Reviewer #1: No

Reviewer #2: Yes

5. Review Comments to the Author

Reviewer #1: In this epidemiologic study the authors collected soil and water samples from several regions (wet, dry and semi-dry) of Sri Lanka based on probable exposure history of confirmed leptospirosis patients and tested it by PCR. There are few leptospirosis epidemiologic studies from Sri Lanka. From the wet area 2/12 water sites were positive. From the dry area 1 well in areas of high human contact were positive. A specific positive site in the dry area was chosen to do serial sample testing from 10 wells surrounding a positive well (nr 9) and they found that at different times 4 to 6 of the 10 wells in that site were positive for Leptospira. The work has merit but as described in this paper is very difficult to follow and the abstract does not capture the essence or importance of the work done. The description of the method to collect the samples including a better flow chart, figures, tables and writing to need to be comprehensively revised.

Other comments:

Methods: If the primers detect nonpathogenic Leptospira species, what is the rationale of detecting virulent species in the water and soil samples?

Study design and presentation of results: Very confusing descriptions of all data.

Throughout the paper (including abstract) avoid using Components 1, 2, 3 and just define the sites as 1-X nr of water and soil samples from Wet area, 2- X nr of Wells in Dry Area, 3 – 10 wells surrounding a positive well in dry area.

Discussion: down the claims of positive Leptospiral transmission routes or correlations with disease hotspots since no clinical data is shown.

It is difficult to design any prophylactic measures based on an epidemiological study. Reservoir based studies or other specific animal study can therefore open such avenues to search for a better alternatives than available Doxycycline.

Reviewer #2: This is a nice manuscript describing findings of Leptospira in the environment, adding to our knowledge of transmission of Leptospira in the environment.

Specific comments:

The authors use "random" to describe Leptospira in the environment, but I'm not sure that's the correct term. Maybe transient? Or variable? Maybe conditions have to be ideal to reach levels above the limit of detection, and with clear correlations of lepto cases in humans and rainy seasons, and as shown in this study, proximity to a lake where feral animals are prevalent, I'm not sure "random" should be used.

Can the authors justify why they tossed the first centrifuged water pellet? It's mentioned in limitations that they may have lost some lepto with the pellet. Also, a DNA blood mini kit was used for environmental water - can the authors justify the extraction kit use, or were they able to determine whether or not PCR inhibitors were present?

Can the authors describe in better detail the definition of PCR positivity? It's not clear to me how many "replicates" were done, and whether the replicates were in the PCR reaction or replicate samples.

What is the difference between an "abundant paddy field" and a paddy field (line 183)? More vegetation? Larger?

It might be helpful in the results section to clarify that the soil sequencing detected both pathogenic and non-pathogenic Leptospira, and that non-pathogenic Leptospira are considered ubiquitous.

6. PLOS authors have the option to publish the peer review history of their article (what does this mean?). If published, this will include your full peer review and any attached files.

Reviewer #1: No

Reviewer #2: No

---

## [Author Response · Author response to Decision Letter 0]

6 Nov 2021

Respond to reviewers document is attached separately.

---

## [Decision Letter · Decision Letter 1]

11 Jan 2022

PONE-D-21-25422R1The variable presence of Leptospira in the environment; An epidemiological explanation based on serial analysis of water samplesPLOS ONE

Dear Dr. Agampodi,

Thank you for submitting your manuscript to PLOS ONE. After careful consideration, we feel that it has merit but does not fully meet PLOS ONE’s publication criteria as it currently stands. Therefore, we invite you to submit a revised version of the manuscript that addresses the points raised during the review process.

The requested edits are minor and, if corrected, will likely result in acceptance for publication.  We look forward to receiving your revised manuscript.

Please include the following items when submitting your revised manuscript:A rebuttal letter that responds to each point raised by the academic editor and reviewer(s). You should upload this letter as a separate file labeled 'Response to Reviewers'.A marked-up copy of your manuscript that highlights changes made to the original version. You should upload this as a separate file labeled 'Revised Manuscript with Track Changes'.An unmarked version of your revised paper without tracked changes. You should upload this as a separate file labeled 'Manuscript'.If applicable, we recommend that you deposit your laboratory protocols in protocols.io to enhance the reproducibility of your results. Protocols.io assigns your protocol its own identifier (DOI) so that it can be cited independently in the future. For instructions see: https://journals.plos.org/plosone/s/submission-guidelines#loc-laboratory-protocols. Additionally, PLOS ONE offers an option for publishing peer-reviewed Lab Protocol articles, which describe protocols hosted on protocols.io. Read more information on sharing protocols at https://plos.org/protocols?utm_medium=editorial-email&utm_source=authorletters&utm_campaign=protocols.

We look forward to receiving your revised manuscript.

Kind regards,

R. Mark Wooten, Ph.D.

Academic Editor

PLOS ONE

Journal Requirements:

Reviewers' comments:

Reviewer's Responses to Questions

**Comments to the Author**

1. If the authors have adequately addressed your comments raised in a previous round of review and you feel that this manuscript is now acceptable for publication, you may indicate that here to bypass the “Comments to the Author” section, enter your conflict of interest statement in the “Confidential to Editor” section, and submit your "Accept" recommendation.

Reviewer #1: All comments have been addressed

Reviewer #2: All comments have been addressed

2. Is the manuscript technically sound, and do the data support the conclusions?

Reviewer #1: Yes

Reviewer #2: Yes

3. Has the statistical analysis been performed appropriately and rigorously? 

Reviewer #1: Yes

Reviewer #2: Yes

4. Have the authors made all data underlying the findings in their manuscript fully available?

Reviewer #1: Yes

Reviewer #2: Yes

5. Is the manuscript presented in an intelligible fashion and written in standard English?

Reviewer #1: Yes

Reviewer #2: Yes

6. Review Comments to the Author

Reviewer #1: Abstract

Line 23: define the classic triad

Methods:

Line 105 and others: Purposive? What does this word mean?

Discussion

Line 279: How much is “a satisfactory concentration of Leptospira needed in the environment”.

Line 309: This concept of intermittent positivity versus repetitive contamination should be mentioned in the abstract.

Reviewer #2: Thank you, the revision provided clarity and I think the manuscript provides valuable information about Leptospira in the environment in an endemic area. I have only one small comment, to de-capitalize "leptospirosis" in the middle of a sentence.

7. PLOS authors have the option to publish the peer review history of their article (what does this mean?). If published, this will include your full peer review and any attached files.

Reviewer #1: No

Reviewer #2: No

---

## [Author Response · Author response to Decision Letter 1]

14 Jan 2022

Comment - Please review your reference list to ensure that it is complete and correct.

Reply - Done

Comment - If you have cited papers that have been retracted, please include the rationale for doing so in the manuscript text, or remove these references and replace them with relevant current references. Any changes to the reference list should be mentioned in the rebuttal letter that accompanies your revised manuscript.

Reply - No retracted papers are cited.

Comment - Abstract

Line 23: define the classic triad

Reply - Thank you very much for the comment. We included the epidemiological triad within brackets in the given place. 

Comment - Methods:

Line 105 and others: Purposive? What does this word mean?

Reply - Thank you very much for the comment. We understand the concern. We used the term ‘purposive’ to indicate the non-probability sampling technique. However, the term purposive is more general.

The term purposive was used in two places to explain the sample selection technique of two components of the study. 

Actually, we selected the sites for the first component based on the probable sites of contamination by leptospirosis patients. These were during our clinical studies, investigators past experiences or the experiences of the physicians.

We selected the sites for the second component considering the possibility of daily exposure by the humans. 

For, better clarity, we changed the sentences as follows.

“For the water samples, water sources were selected purposefully based on the probable sites of contamination of diagnosed leptospirosis patients. Sampling was conducted at 12 sites representing all three climatic zones: dry, wet, and intermediate.”

“For the second component, water collection sites were selected purposefully considering the possibility of daily human contacts.”

Comment - Discussion

Line 279: How much is “a satisfactory concentration of Leptospira needed in the environment”.

Reply - I humbly thank you for this comment. This is something we can’t give an exact answer and this depends on several factors as shown in the last figure. 

However, we needed to explain that the higher concentration due to recent animal contamination could be a factor associated with this. 

Further studies on satisfactory concentration could be an interesting research area to explore. 

We added the following sentence at the end of the paragraph.

“Studies on required minimal leptospira concentration in the environment could be new research area to be explored.”

Comment - Line 309: This concept of intermittent positivity versus repetitive contamination should be mentioned in the abstract.

Reply - Thank you very much for the nice suggestion. We added the following sentence to the abstract.

“This intermittent nature of positivity could be explained by the repetitive contamination by animal urine.”

Comment - Thank you, the revision provided clarity and I think the manuscript provides valuable information about Leptospira in the environment in an endemic area. I have only one small comment, to de-capitalize "leptospirosis" in the middle of a sentence

Reply - Thank you very much. We did the changes in the manuscript.

---

## [Editor Report · Decision Letter 2]

26 Jan 2022

The variable presence of Leptospira in the environment; An epidemiological explanation based on serial analysis of water samples

PONE-D-21-25422R2

Dear Dr. Agampodi,

We’re pleased to inform you that your manuscript has been judged scientifically suitable for publication and will be formally accepted for publication once it meets all outstanding technical requirements.

Kind regards,

R. Mark Wooten, Ph.D.

Academic Editor

PLOS ONE
---

## [Editor Report · Acceptance letter]

7 Feb 2022

PONE-D-21-25422R2 

The variable presence of *Leptospira* in the environment; An epidemiological explanation based on serial analysis of water samples 

Dear Dr. Agampodi:

I'm pleased to inform you that your manuscript has been deemed suitable for publication in PLOS ONE. Congratulations! Your manuscript is now with our production department. 

Kind regards, 

on behalf of

Dr. R. Mark Wooten 

Academic Editor

PLOS ONE